# Factors Influencing the Acceptance of COVID-19 Vaccines in a Country with a High Vaccination Rate

**DOI:** 10.3390/vaccines10050681

**Published:** 2022-04-25

**Authors:** Daniela Toro-Ascuy, Nicolás Cifuentes-Muñoz, Andrea Avaria, Camila Pereira-Montecinos, Gilena Cruzat, Katherine Peralta-Arancibia, Francisco Zorondo-Rodríguez, Loreto F. Fuenzalida

**Affiliations:** 1Facultad de Ciencias de la Salud, Instituto de Ciencias Biomédicas, Universidad Autónoma de Chile, Santiago 8910060, Chile; daniela.toro@uautonoma.cl (D.T.-A.); nic.cifuentes@gmail.com (N.C.-M.); camila.pereira1@cloud.uautonoma.cl (C.P.-M.); gilena.cruzat@cloud.uautonoma.cl (G.C.); katherine.peralta@cloud.uautonoma.cl (K.P.-A.); 2Facultad de Ciencias Sociales y Humanidades, Universidad Autónoma de Chile, Santiago 8910132, Chile; andrea.avaria@uautonoma.cl; 3Departamento de Gestión Agraria, Facultad Tecnológica, Universidad de Santiago de Chile, Santiago 9170125, Chile

**Keywords:** COVID-19 vaccine acceptance, vaccine confidence, risk perception

## Abstract

Control of the COVID-19 pandemic largely depends on the effectiveness of the vaccination process. An understanding of the factors that underlie the willingness to accept vaccination contributes pivotal information to controlling the pandemic. We analyzed the association between the willingness to accept the available COVID-19 vaccines and vaccine determinants amidst the Chilean vaccination process. Individual-level survey data was collected from 744 nationally representative respondents and multivariate regression models were used to estimate the association between outcome and explanatory variables. We found that trust in COVID-19 vaccines, scientists, and medical professionals significantly increased the willingness to: accept the vaccines and booster doses, as well as annual vaccinations and the vaccination of children. Our results are critical to understanding the acceptance of COVID-19 vaccines in the context of a country with one of the world’s highest vaccination rates. We provide useful information for decision-making and policy design, in addition to establishing guidelines regarding how to effectively explain vaccination programs to citizens.

## 1. Introduction

The sudden appearance and spread of the severe acute respiratory syndrome coronavirus 2 (SARS-CoV-2) in the human population in 2019 has had catastrophic consequences, with global deaths reaching over 6.1 million worldwide [1,2]. In March 2020, the World Health Organization declared the coronavirus disease 2019 (COVID-19) a pandemic. Worldwide, non-pharmaceutical interventions, such as mask use, lockdowns, and social distancing helped to slow the pandemic down until vaccines became available. The remarkably rapid development of vaccines against SARS-CoV-2 is turning COVID-19 into a preventable disease [3]. However, several challenges regarding the COVID-19 vaccination process must still be addressed, including the reluctance to accept the vaccines.

The success of the vaccination process strongly depends upon underlying social factors, mainly the willingness to accept the vaccination, trust in stakeholders related to the vaccination, vaccine-specific factors, communication and media, historical influences, religion, gender, socioeconomic status, politics, geographic barriers, prior experience with vaccinations, risk perception, and design of the vaccination program [4]. Some studies have utilized surveys to explore the acceptance of COVID-19 vaccines in different countries, including the United States [5,6,7], the United Kingdom [7,8,9], China [10,11], Indonesia [12], Italy [13], Ireland [8], and Japan [14]. In addition, surveys have explored vaccine acceptance in groups of European countries [15], Arab countries [16,17], and other countries worldwide [18,19,20]. Several of these studies have concluded that the willingness to accept a COVID-19 vaccine differs depending on the age, educational and economic level, credibility of government decisions and the perceived risk of the COVID-19 [5,10,17,20,21,22]. In a global vaccine study carried out in 19 countries, responses were reported to have been highly heterogeneous, depending on the country surveyed [20]; therefore, it is important to understand the acceptance of a vaccine in the context of specific countries or regions [22].

The first reported case of COVID-19 in Chile occurred in March 2020. Community transmission of the virus caused the peak of the first wave of COVID-19 cases in June 2020 [23]. Non-pharmaceutical interventions were key to slow down the pandemic in Chile and prevent further deaths; as of April 2022, a total of 56,000 people had died in Chile due to COVID-19. The first vaccine approved for emergency use in Chile was the Pfizer (BNT162b2) on 16 December 2020. The second vaccine approved by the Chilean authorities for emergency use, was the virion-inactivated CoronaVac vaccine, on 20 January 2021. CoronaVac has been the most widely used vaccine in Chile, with over 26 million doses administered, followed by Pfizer, with over 21 million doses allocated by early April 2022 [24]. Studies have shown a high effectiveness of CoronaVac in preventing symptomatic COVID-19, hospitalizations and death [25]. Shortly after the approval of the CoronaVac vaccine, the emergency use authorization of the Oxford-AstraZeneca (ChAdOx1), Johnson & Johnson (Ad26.COV2), CanSino (Ad5-nCoV) and Sputnik V (GAM-COVIDVac) vaccines was also granted. Surprisingly, since the beginning of the COVID-19 vaccination campaign, Chile has emerged as one of the most successful countries worldwide regarding the number of vaccine doses administered per person [26,27,28], outcompeting first-world countries in this regard. Approximately 92% of the Chilean population was fully vaccinated (booster doses included) in April 2022, placing the country within the five most successful countries in terms of vaccine doses administered per person in the world [26]. These numbers contrast strikingly with the large percentages of adult populations that remain unvaccinated and/or unwilling to receive vaccinations in first world countries: 19% in Germany, 18% in the Netherlands, 17% in the UK, 17% in France, and 12% in Canada, to name a few examples [26]. In this context, vaccination against SARS-CoV-2 remains a serious challenge for most countries worldwide, although for different reasons: in countries with low and middle income (LMIC) there is a higher willingness to accept a vaccine, but limited access to them, while in countries with higher incomes there is ample access to vaccines, but high levels of hesitancy toward receiving a vaccination [29]. A small high-income country like Chile, with broad vaccination coverage, can offer key lessons in terms of how to address the challenges regarding the COVID-19 vaccination process. Understanding the determinants of vaccine acceptance is key for decision-making, establishing diverse strategies according to the characteristics and social determinations of the population, and identifying the subjectivities that underlie the decision to vaccinate.

To obtain scientific evidence amidst the Chilean vaccination campaign, we analyzed the association between the level of willingness to accept a COVID-19 vaccine and determinants of the vaccines in Chile. We focused on how trust in vaccines, stakeholders, and people’s perceptions regarding the effectiveness of prevention practices, the risk of infection, and possible side effects of vaccines were related to people’s willingness to accept a SARS-CoV-2 vaccine, a booster dose, an annual vaccination, and the vaccination of children. Socio-demographic variables associated with the acceptance of the vaccination process were also evaluated. Our study identified at least two key aspects that contribute to vaccine acceptance in the Chilean adult population: a) A high trust in scientists and health workers, as well as a moderate trust in the media; b) a high perception of risk of infection. Our results aid in identifying the subjective dimensions related to the population’s decision to be vaccinated, offering keystone evidence that could help other countries face this pandemic. The information provided by our study could also help improve public health communication strategies.

## 2. Materials and Methods

Ethical approval for this study was obtained from the Universidad Autónoma de Chile ethics committee on 19 April 2021 with reference to CEC 10-21. The study utilized a cross-sectional online survey design of adults living in Chile. A validation process was carried out to help ensure the content validity, face validity, criterion-related validity, construct validity, reliability, and intelligibility of the questions and the overall questionnaire. First, the content and face validity were analyzed by three experts in immunology and public health. Second, criterion-related validity was evaluated by experts in design and preparation of questionnaires related to public health research. Third, the construct validity was confirmed through factor analyses and tests of Cronbach’s Alpha of the different items included in the questionnaire. Fourth, the reliability of questions was ensured using established measures or scales that have proved to be valid in previous social research [30]. Finally, the questions and their arrangement in the questionnaire were tested and adjusted to assess their intelligibility in a different sample of individuals from the final sample used in the study.

The structured questionnaire was applied through an online platform between May and June 2021. During this period, the study captured the attitudes of people toward Chile’s first efforts to control COVID-19 through vaccination, offering insights as to how to face this challenging task worldwide. Considering that our sampling design was based on an online survey, it is difficult to ensure that our survey represented a more-general population. Despite this weakness, we are also strongly aware that our survey and study have many strengths that compensate for this weakness [30]. For instance, our online survey was able to capture the opinions of a variety of individuals from different administrative regions of Chile, mainly when the country was under strict lockdowns, prohibition of social meetings, quarantines, and other societal restrictions to control COVID-19. To avoid selection biases, we shared the online questionnaire on diverse social medias and groups of people from different Chilean regions. We tried to ensure the balance between respondents from the Metropolitan Region (which concentrates 40% of the national population) and other regions. In fact, a total of 744 volunteers completed the questionnaire, with 58% distributed throughout the Metropolitan Region and 42% throughout the rest of the country. Considering that the Chilean adult population (more than 18 years old) includes around 12 million people, our sample size could be representative of this population, with a confidence level of 95% and a margin of error of 3.5%.

The questionnaire captured variables regarding the willingness to accept vaccines, variables of trust, variables of perceptions, and socio-demographic variables, which were grouped as (a) outcome, (b) explanatory, and (c) control variables. Outcome variables, four in total, were related to the willingness to accept vaccines: (i) The willingness to accept SARS-CoV-2 vaccines was estimated with three values (3 = yes, 2 = maybe, and 1 = no); (ii) The willingness to accept a vaccine booster shot after having a complete vaccination scheme was included as a dichotomous variables (yes = 1); (iii) The willingness to accept an annual vaccination was assessed on a 4-value ordinal scale (from 1 = not willing to 4 = highly willing) and; (iv) The willingness to accept the vaccination of children was assessed on a 4-value ordinal scale (from 1 = definitively no to 4 = definitely yes). 

Explanatory variables, also four in total, focused on trust in (i) vaccines, (ii) stakeholders, (iii) social media, and (iv) press. To reflect the trust in vaccines, our aim was to prompt people to report their trust in each of the vaccines approved in Chile: CoronaVac, Pfizer (BNT162b2), CanSino (Ad5-nCoV), Oxford-AstraZeneca (ChAdOx1), Sputnik V Gam-COVID-Vac), and Johnson& Johnson (Ad26.COV2). We also assessed the trust in different stakeholders, including scientists, medical professionals, professionals of the Chilean Public Health Institute (ISP), professionals of the World Health Organization (WHO), Ministry of Health, family, friends, politicians, and religious leaders. Similarly, we included questions to assess trust in social media as sources of information, which could influence the willingness to accept vaccines and treatments. Among social media, we asked people to express their trust in general websites, Twitter, WhatsApp, Facebook, Instagram, and Tik Tok. Lastly, trust in the press was also included in the questionnaire as formal information sources that may influence individuals, such as national and international newscasts, national and international newspapers, and radio broadcasts. For all of these questions, we used a 4-value ordinal scale, from 1 = no trust to 4 = high trust. 

We prompted people to assess their perceptions about a set of factors that might explain their level of willingness to accept the vaccination process against COVID-19. First, individuals’ perception of the effectiveness of COVID-19 prevention practices was captured through a 4-value ordinal scale, from 1 = not effective to 4 = highly effective. We included the most frequent Chilean prevention practices, such as vaccination, use of mask, hand-washing, social distancing, avoiding social gatherings, quarantine, and “sanitary clinics”, which refer to free Chilean establishments that meet the quality and safety conditions of both the person who must comply with the quarantine or isolation measure and the staff in charge of their care, available to people positive with COVID-19 or those that must undergo a quarantine due to close contact with an infected individual. Second, we also captured the perceived risk of infection on an ordinal scale with four values (from 1 = not probable to 4 = highly probable). Third, the preoccupation regarding side effects of the vaccines was assessed by a self-reported level of concern using a scale from not worried (=1) to highly worried (=4). Fourth, we also assessed the perceived comprehension of the CoronaVac, Pfizer, CanSino, Oxford-AstraZeneca, Sputnik V, and Johnson& Johnson vaccines, where individuals reported the level of information they considered having about each vaccine. The individual self-report was assessed using a 4-value ordinal scale (from 1 = no information to 4 = a lot of information). Fifth, the idea that prevention practices could be relaxed due to vaccination was proxied with the claim “the vaccination” (i) allows me to stop using a mask, (ii) allows me to relax the measures I take to prevent physical interactions with other people, (iii) will prevent me from contracting COVID-19, (iv) will prevent me from getting severely ill, and (v) will stop the COVID-19 pandemic. The answer to each practice was captured on a 5-value ordinal scale (from 1 = highly disagree to 5 = highly agree). Sixth, we evaluated the perceived impact of the COVID-19 pandemic on quality of life. We proxied the concept of quality of life by evaluating the pandemic’s impacts on job, education, health, familial coexistence, and general well-being of the household members. We assessed the impact of the pandemic on a 5-value ordinal scale, from 1 = very negative to 5 = very positive. 

Finally, a set of questions was included to capture control variables that might be associated with the outcome or explanatory variables. We asked whether the respondent or any family member had been infected with COVID-19, and whether the illness was acute. We also collected information on age, gender, administrative region of residence, schooling, and nationality.

### Statistical Analysis

We carried out factor analyses in order to estimate the retained factors included in the explanatory variables of trust and perception, while Cronbach’s Alpha was estimated to measure the internal consistency and reliability among the set of retained factors of each explanatory variable. Factors included in each variable of trust and perception were explained above, and we provide the results in detail before explaining the results of multivariate regressions. We took the average of retained factors to create the explanatory variables of trust and perceptions included in the multivariate regression analyses.

We were interested in estimating the association between outcome variables of willingness to accept vaccination and explanatory variables of trust and perceptions, while controlling for age, gender, and schooling. We used the following general Equation (1):WTA = α Trust + β Perception + γ Control(1)
where WTA represents one of the four outcome variables of willingness to accept. Trust includes the explanatory variables of trust, while Perception stands for variables of perceptions. Lastly, Control captures the control variables including (a) age, (b) gender, (c) schooling, (d) having contracted COVID-19, (e) sub-national region of residence, and (f) nationality. The lack of collinearity among explanatory and control variables was checked to adjust the multivariate regression models.

We computed the odds ratio, represented as α, β, and γ in the Equation (1), as a measure of association of outcome variables with the set of explanatory variables of trust and perception, as well as with the control variables. The odds ratio is well understood as an effect size measure for logistic regression models [31]. The odds ratio represents the ratio of the odds that an outcome variable will occur given an explanatory variable compared to the odds of the outcome occurring in the absence of the explanatory variable. If an odds ratio is greater than 1, then an explanatory variable induces a higher level of acceptance, relative to the control of other variables used in the model. On the other hand, an odds ratio less than 1 suggests that an explanatory variable influences a lower willingness. We described only those results where the 95% confidence intervals excluded zero, which were deemed statistically credible. Although our models did not measure causal effects, log cumulative odds ratios showed how the variables related to willingness responded to variables of perception and trust or how the associations varied among genders or age groups. The access to self-reported perceptions provided correlational evidence regarding which factors explained a greater willingness to accept the SARS-CoV-2 vaccination process.

We used ordered logistic multivariate models when the outcome variable had a meaningful order in more than two categories, such as the variables of willingness to accept a SARS-CoV-2 vaccine, annual vaccines, and vaccination of children. In turn, logistic multivariate models were used for analyzing a dichotomous outcome variable, such as the willingness to accept a COVID-19 vaccine booster shot. In all of the models, we used the same explanatory and control variables. For each outcome variable, we adjusted a set of different models omitting one or more explanatory and control variables. We used the Akaike information criterion (AIC) to perform model comparisons and select the model with the best goodness of fit and parsimony. We ranked the AIC values and defined that the lowest AIC value represents the best fitted model (Appendix A).

## 3. Results

Between 21 May and 21 June of 2021, a total of 744 adults in Chile were interviewed via online surveys. The self-administered questionnaire was distributed through social networks. A summary of the socio-demographic characteristics of the respondents included in this study is shown in Appendix A. The questions were aimed at estimating four outcome variables related to the willingness to accept: (i) a SARS-CoV-2 vaccination (0 = not, 1 = maybe, 2 = yes), (ii) a vaccine booster dose (1 = yes), (iii) an annual vaccination (1 = not willing to 4 = highly willing), and (iv) the vaccination of children (1 = definitively no to 4 = definitely yes). The questionnaire also included, as explanatory variables, a set of variables of trust and perceptions associated with the vaccination process. These questions were aimed at describing the perception of risk and trust, and responses were also recorded on a 3 or 4-point ordinal scale of agreement or disagreement. For instance, trust in the different COVID-19 vaccine stakeholders was estimated using the scale: “No trust”, “Little trust”, “Some trust”, and “High trust”. The full questionnaire is shown in the Appendix A. In addition, Appendix A indicates milestones that occurred during the data collection period, according to the development of the pandemic and vaccination process in Chile.

Most of the respondents (93.4% *n* = 695) had received at least one dose of a SARS-CoV-2 vaccine at the time of the survey (mainly CoronaVac and Pfizer), whereas 3.9% (*n* = 29) had not yet decided if they would accept a SARS-CoV-2 vaccine and 2.7% (*n* = 20) affirmed they would definitely not accept a SARS-CoV-2 vaccine. When asked if they would accept a hypothetical booster dose, 88.2% (*n* = 656) of the respondents reported that they would accept and 57.8% (*n* = 430) affirmed they would definitively accept a yearly vaccination if necessary, similar to the vaccine schedule for the influenza virus. When asked if, in the case of having children under 16 years old, they would accept that their children could be vaccinated against SARS-CoV-2, 62.5% (*n* = 175) reported that they would “definitively accept” a SARS-CoV-2 vaccine for their children.

Multivariate regression models were used to estimate the association between the outcome and explanatory variables. An ordered logistic regression model was adjusted to analyze the outcome variables related to the willingness to accept a SARS-CoV-2 vaccine, an annual vaccination, and the vaccination of children, while a logistic regression model was used to analyze the willingness to accept a booster dose. Besides estimating the associations among the entire sample of individuals (*n* = 744), we also analyzed if factors varied their associations when comparing samples of men (*n* = 260) and women (*n* = 484), as well as samples of young adults (18–29 years old, *n* = 206) and adults (30–59, *n* = 503). We excluded the sample of elderly people (>59 years old) only in this analysis because of the small sample (*n* = 35). For each outcome variable, we selected the model with the best goodness of fit and parsimony using the Akaike information criterion (AIC) (Appendix A).

### 3.1. Trust in SARS-CoV-2 Vaccines Increased the Willingness to Accept a SARS-CoV-2 Vaccine, Booster Dose, Annual Vaccination, and Vaccination of Children

People’s trust varied in relation to the vaccine in question (Kruskal–Wallis test: Chi-squared = 509, d.f. = 5, *p* < 0.001) (Figure 1A). While the reported trust did not differ between the CoronaVac and Pfizer vaccines (Dunn’s test with Bonferroni adjustment: z = −1.77, *p* = 0.57), the trust in both the Pfizer and CoronaVac vaccines was significantly greater than that in the other four vaccines approved by Chilean authorities (Dunn’s test with Bonferroni adjustment between Pfizer vs. CanSino z = 14.1, *p* < 0.001; vs. AstraZeneca z = 14.6, *p* < 0.001; vs. Sputnik V z = 16.1, *p* < 0.001; vs. Johnson& Johnson z = 14.6, *p* < 0.001); and between CoronaVac vs. CanSino z = 12.5, *p* < 0.001; vs. AstraZeneca z = 12.9, *p* < 0.001; vs. Sputnik z = 14.5, *p* < 0.001; vs. Johnson z = 13.0, *p* < 0.001) (Figure 1A).

There were no differences in the reported trust in the AstraZeneca, CanSino, Sputnik V, and Johnson & Johnson vaccines. In spite of the differences found in the trust of the aforementioned vaccines, trust in all of the vaccines was retained in one factor (Factor Analysis: Eigenvalue of Factor 1 = 3.6; LR test: chi-squared = 2032.1, *p* < 0.001), with a very high reliability coefficient (Cronbach’s alpha = 0.89). We therefore took the average of the trust in all of the vaccines to create a variable of overall trust in vaccines against SARS-CoV-2 and used it in the multivariate regression models.

Multivariate models suggest that the increase of one unit value in trust of SARS-CoV-2 vaccines increased 4.1 times the willingness to accept SARS-CoV-2 vaccines (95%CI = 2.0–8.2, *p* < 0.001), 3.2 times the willingness to accept a booster dose (95%CI = 1.8–5.6, *p* < 0.001), twice the willingness to accept an annual vaccination (95%CI = 1.6–2.8, *p* < 0.001), and 1.9 times the willingness to vaccinate children (95%CI = 1.4–2.6, *p* < 0.001) (Table 1, row (a)). When comparing results between genders, women showed significant associations among all of the willingness variables with trust in SARS-CoV-2 vaccines (Appendix A). On the other hand, men showed a significant association between the willingness to accept the annual vaccination and trust in SARS-CoV-2 vaccines (Appendix A). Interestingly, it was observed, in both young adults and adults, that an increase of trust in SARS-CoV-2 vaccines induced a higher willingness to receive a SARS-CoV-2 vaccination, booster dose, annual vaccination, and the vaccination of children (Appendix A).

### 3.2. Trust in Scientists and Medical Professionals Increased the Acceptance of a SARS-CoV-2 Vaccination, Booster Dose, Annual Vaccination, and the Vaccination of Children, While Trust in Religious Leaders Reduced the Willingness to Accept an Annual Vaccination and the Vaccination of Children

People’s trust in stakeholders varied significantly (Kruskal–Wallis test: 2285.9, d.f. = 8, *p* < 0.001) (Figure 1B). Scientists received the highest score of trust among all of the stakeholders included in the study (Dunn’s test with a Bonferroni adjustment: z > 6.7 and *p* < 0.001 in all comparisons), followed by medical professionals, the Chilean Public Health Institute (ISP), and WHO professionals (Figure 1B). The lowest scores of trust were reported for politicians and religious leaders. For instance, 63% (*n* = 472) and 28% (*n* = 207) of individuals reported a “high trust” and “some trust”, respectively, in scientists, and more than 70% reported a “high trust” (43%, *n* = 323) or “some trust” (36%, *n* = 267) in medical professionals. On the contrary, 64% (*n* = 479) and 45% (*n* = 334) of the surveyed individuals reported not trusting religious leaders and politicians (Figure 1B). Factor analysis suggests that the variability of trust in stakeholders included in the study can be explained by four groups: (a) the scientific and medical professional group, including, WHO, and ISP professionals (Retained Factor 1: Eigenvalue = 2.3, LR test Chi-square = 1407, *p* < 0.001, Cronbach’s alpha = 0.85); (b) the politicians group, including the authorities of the Ministry of Health (Retained Factor 1, Eigenvalue = 0.8; LR test Chi-square = 247.3, *p* < 0.001; Cronbach’s alpha = 0.7); (c) the relative group, including family and friends (Retained Factor 1 = 0.8, LR test Chi-Square = 289.9, *p* < 0.001; Cronbach’s alpha = 0.7), and (d) a fourth group with religious leaders only. To incorporate trust in stakeholder groups into the multivariate regression models, we averaged the reported trust scores for all of the stakeholders included in each group.

We found evidence that individuals with higher trust in scientists and medical professionals significantly increased (by 2.4 times) their acceptance of SARS-CoV-2 vaccinations (95% CI = 1.2–5.0, *p* = 0.01), as well as their willingness to accept a booster dose (by 2.8 times) (95% CI = 1.5–5.0, *p* = 0.001) (Table 1, row (b) of the columns (1) and (2)). Similarly, an increase of trust in scientists and medical professionals also increased the willingness to accept an annual vaccination, 2.2-fold (95% CI = 1.6–3.1, *p* < 0.001) and the vaccination of children, 2.6 times (95% CI = 1.8–3.6, *p* < 0.001) (Table 1, row (b) of the columns (3) and (4)). Interestingly, some groups responded differently in relation to their trust in scientists and medical professionals and thus showed comparatively different associations with the willingness variables. For instance, women did not vary their willingness to accept a SARS-CoV-2 vaccine and booster dose when they reported a higher or lower level of trust in scientists and medical professionals. In contrast, men showed that a higher trust in scientists and medical professionals increased 46.1 and 4.2 times their willingness to accept a SARS-CoV-2 vaccine (95% CI = 2.5–862.1, *p* = 0.01) and booster dose (95% CI = 1.3–13.1, *p* = 0.02), respectively (Appendix A). In the case of the willingness to accept an annual vaccination and the vaccination of children, both women and men showed a similar positive impact of trust in scientists and medical professionals (Appendix A). We found evidence that young people and adults differed in terms of how their trust in scientists and medical professionals impacted the variables of willingness. For instance, young people did not vary their willingness to accept a SARS-CoV-2 vaccine, booster dose, annual vaccination, and the vaccination of children as their levels of trust in scientists and medical professionals increased (Appendix A). On the contrary, adults showed that a higher level of trust increased their willingness to receive a SARS-CoV-2 vaccine, booster dose, annual vaccination, and the vaccination of children by around three-fold (Appendix A). In contrast, our results showed that willingness to accept both an annual vaccination (95%CI = 0.6–0.9, *p* = 0.02) and the vaccination of children (95%CI = 0.6–0.9, *p* = 0.004) decreased by 30% with a one unit increase of trust in religious leaders (Table 1, row (d) of the columns (3) and (4)). However, when comparing between genders, the trust in religious leaders decreased the willingness to accept the booster dose only among men. Both women and men decreased their willingness to vaccinate children as their trust in religious leaders increased (Appendix A). Trust in religious leaders impacted the willingness scores differently between age groups. For instance, only adults showed a decrease in their willingness to accept a booster dose as trust in religious leaders increased, while only young individuals with greater trust in religious leaders decreased their willingness to accept an annual vaccination and the vaccination of children. Lastly, people’s trust varied among social media (Figure 1C). Multivariate analyses showed that an increase of trust in social media was associated with a lower willingness to accept a booster dose (OR = 0.4, 95%CI = 0.2–0.7, *p* = 0.001) and the vaccination of children (OR = 0.7, 95%CI = 0.7–1.0, *p* = 0.03) (Table 1, row (f) of the columns (2) and (4)).

### 3.3. A Higher Perceived Risk of Infection and Effectiveness of Prevention Practices as Well as Less Concern Regarding Side Effects of Vaccines Increased the Willingness to Accept a SARS-CoV-2 Vaccine

People’s perceptions regarding the effectiveness of prevention practices varied according to the practice (Kruskal–Wallis test: Chi-squared = 591.1, d.f. = 7, *p* < 0.001) (Figure 1D). For instance, the vaccination was perceived as more effective compared to lockdown (Dunn’s test with Bonferroni adjustment: z = 10.2, *p* < 0.001), but less effective than the use of a mask (z = −6.22, *p* < 0.001), hand-washing (z = −10.1, *p* < 0.001), physical distance (z = −9.3, *p* < 0.001), avoidance of social gatherings (z = −7.1, *p* < 0.001), and quarantine (z = −3.9, *p* = 0.001). The use of a mask was perceived as less effective than hand-washing (z = −4.8, *p* < 0.001) and physical distance (z = −3.3, *p* = 0.011). The lockdown was perceived as the least effective practice to prevent the infection of SARS-CoV-2 among all of the practices included in the study (Figure 1D). Notwithstanding the different perceptions of effectiveness across practices, results suggest that the variability of perceptions can be retained in one factor (factor analysis: eigenvalue of factor 1 = 3.0; LR test: chi-squared = 1292.1, *p* < 0.001), with the set of variables showing a high internal consistency (Cronbach’s alpha = 0.81). We calculated the overall perceived effectiveness of the prevention practices for each individual as the average value of perceived effectiveness among all practices, and used this new variable in the multivariate analysis.

Most people perceived that the probability of becoming infected with COVID-19 was “little probable” (*n* = 423, 56.8%) or “some probable” (*n* = 239, 32.1%) (Figure 1E). Only 49 of the surveyed individuals (6.6%) reported that getting COVID-19 was “highly probable”. People also reported being “little worried” (*n* = 306, 41%) or “not worried” (*n* = 190, 25.5%) about the side effects of vaccines. A total of only 79 individuals from the sample reported being “highly worried” about the side effects of vaccines (*n* = 79, 10.6%) (Figure 1E). People’s trust in diverse types of press media proved to vary greatly (Figure 1F).

When the perceived effectiveness of infection prevention practices was a score unit higher, the willingness to accept a SARS-CoV-2 vaccine increased two-fold (95%CI = 1.0–4.5, *p* = 0.05), the booster dose by 2.4 times (95%CI = 1.3–4.5, *p* = 0.005), an annual vaccination by 2.4 times (95%CI = 1.6–3.4, *p* < 0.001), and the vaccination of children 2.4-fold (95%CI = 1.6–3.5, *p* < 0.001) (Table 1, row (h)). Moreover, when the perception of infection risk increased by one unit value, the willingness to accept a SARS-CoV-2 vaccine doubled (95%CI= 1.1–3.7; *p* = 0.02) and the willingness to accept an annual vaccination increased by 1.4 times (95%CI = 1.1–1.8; *p* = 0.01) (Table 1, row (i) of the columns (1) and (3)). In contrast, when preoccupation regarding the side effects of SARS-CoV-2 vaccines increased by one unit value, the willingness to accept a SARS-CoV-2 vaccine decreased by 40% (95% CI = 0.4–0.9; *p* = 0.01) and the willingness to vaccinate children decreased by 20% (95% CI = 0.7–1.0; *p* = 0.02) (Table 1, row (j) of the columns (1) and (4)).

Moreover, we found that associations of willingness variables with the perceived effectiveness of prevention practices, perceived infection risk and preoccupation regarding side effects, varied among genders and cohorts. For instance, when comparing results between genders, women showed significant associations among all willingness variables with the perceived effectiveness of prevention practices (Appendix A). On the other hand, men showed a significant association between the willingness to accept the annual vaccination and the perceived effectiveness of prevention practices (Appendix A). Analyses across age groups suggest that only adults increased their willingness to accept a booster dose as their perceived effectiveness of prevention practices increased. Both adults and young people with a higher perceived effectiveness of prevention practices showed a greater willingness to accept an annual vaccination and the vaccination of children (Appendix A). Moreover, perception of infection risk was positively and significantly associated with the willingness to accept a SARS-CoV-2 vaccine (OR = 2.4; 95%CI = 1.1–5.1; *p* = 0.02) and a booster dose (OR = 1.7; 95%CI = 0.9–3.1; *p* = 0.08) among women, but not among men (Appendix A). Preoccupation regarding the side effects of vaccines decreased the willingness to accept SARS-CoV-2 vaccines in both women (OR = 0.5; 95% CI = 0.3–0.8, *p* = 0.008) and men (OR = 0.2; 95%CI = 0.0–0.9; *p* = 0.03), but only men decreased their willingness to vaccinate children as their preoccupation regarding side effects increased (OR = 0.7; 95%CI = 0.5–0.9; *p* = 0.02) (Appendix A). Similarly, increased worry about side effects decreased the willingness to accept a SARS-CoV-2 vaccine (OR = 0.3, 95%CI = 0.1–0.6, *p* = 0.002) and booster dose (OR = 0.3, 95%CI = 0.1–0.8, *p* = 0.01) only among young people. Instead, adults showed that a higher preoccupation regarding side effects decreased their willingness to vaccinate children (OR = 0.7, 95%CI =0.6–0.9, *p* = 0.01) (Appendix A).

Interestingly, when individuals perceived that vaccines reduce the severity of COVID-19, their willingness to vaccinate their children also increased by 20% (OR = 0.8, 95%CI = 0.7–0.9, *p* = 0.002) (Table 1, row (l) of the column (4)). When respondents showed an increased perception that prevention practices could be relaxed owing to vaccination, a reduction in the willingness to accept an annual vaccination was observed (OR = 0.7, 95%CI = 0.6–0.9, *p* = 0.001) (Table 1, row (m) of the columns (3)). Individuals who reported a higher agreement that vaccination would stop the pandemic showed a higher willingness to accept a booster dose (OR = 1.4, 95%CI = 1.0–1.9, *p* = 0.03), an annual vaccination (OR = 1.4, 95%CI = 1.2–1.7, *p* < 0.001), and the vaccination of children (OR = 1.3, 95%CI = 1.1–1.5, *p* = 0.001) (Table 1, row (n) of the columns (2) and (4)). Lastly, individuals who reported that the pandemic positively impacted their subjective well-being showed less willingness to accept a SARS-CoV-2 vaccine (OR = 0.6, 95%CI = 0.3–1.0, *p* = 0.04), but did not vary their willingness to accept a booster dose, annual vaccination, and the vaccination of children (Table 1, row (o)).

## 4. Discussion

We contribute to the existing literature focusing on whether the willingness to accept a vaccination is affected by trust and perceptions of diverse issues related to the COVID-19 pandemic, in the context of a country with one of the world’s highest vaccination rates. Our findings provide relevant information for decision-making processes, particularly for the design and communication of vaccination implementation programs in countries where COVID-19 vaccination remains low.

Our results indicate that higher acceptance of not only SARS-CoV-2 vaccines, but also a booster dose, an annual vaccination, and the vaccination of children, correlates with a high level of trust in the experts in the field, namely scientists and medical professionals. These significant and positive associations emphasize the pivotal role that trust in these experts plays within the vaccination process against COVID-19. In contrast, trust in political or religious leaders proved to be extremely low. In fact, a high trust in these leaders was often associated with a higher refusal to vaccinate. In this scenario, a positive contribution to vaccination could be achieved through partnerships between religious and medical communities [32,33]. Our results are consistent with other studies, showing that it is the high trust in health workers which has been associated with high acceptance of vaccination [29,34,35,36]. These findings are also consistent with the association found between the lack of trust in vaccine experts, and support toward political stances against vaccines [37]. Our findings emphasize how crucial trusted information sources are, suggesting that confidence in health authorities significantly affects vaccine acceptance [38].

Trust in the government has also been associated with vaccine acceptance [39]. In the Chilean context, the government used local health structures to stimulate and bring COVID 19′s vaccination closer to the population as a strategy to boost up the vaccination. However, trust in national and health authorities was lower [40] than in other countries [41]. Trust in the Chilean government could have diminished when COVID-19 continued to increase despite the success of the vaccination campaign (first doses). The lack of trust in the government could be explained by the initial message of the vaccination campaign provided by Chilean authorities, which presented the vaccine as the one of the most effective strategies to control the virus [40]. The low trust in the Chilean authorities and its negative impact on the willingness to accept a vaccination could be the result of generalized criticisms to the authorities due to the management of social inequalities related to how the COVID-19 pandemic impacted socioeconomic groups differently [42]. Our results show the need to further investigate this aspect in future studies, as well as considering the emergence of “new actors” within society in which trust could be deposited, as our findings show: in the scientific world and medical professionals.

Our study identified that adults had a higher acceptance of the vaccination process. Higher acceptance of a vaccine in the adult population could be associated with the perception of the risk of infection of SARS-CoV-2. It has been described that the willingness to be vaccinated is more related to age than to gender, indicating that those who perceive a greater susceptibility to the effects of the virus are more open to accept vaccination and even willing to take a booster dose [41]. Furthermore, the willingness to accept a vaccine in adults has been reported to be considerably higher in LMIC than in countries such as the USA or Russia [29]. Willingness to accept a vaccine was found to be directly related to the perceived personal protective effect of vaccination in adults in LMIC [29]. A cross-sectional study performed in older adults in Italy found that willingness to vaccinate was positively related to a higher educational level [43]. Interestingly, this same study found that the introduction of a green-pass was inversely related to the willingness to accept vaccination. This green-pass allows vaccinated people to use public services and participate in recreational activities such as cinemas, restaurants, sporting events, indoor venues, etc. The introduction of a green-pass in the Chilean vaccination campaign occurred on 14 July 2021. Our study took place before the introduction of the green-pass by national health authorities. Hence, no influence of the green-pass can be associated with our findings, or to the high vaccination rates reached by the vaccination campaign before July 2021. Additional studies have found that vaccinated people think that a green-pass is useful, while unvaccinated individuals think the opposite [44].

In concordance with previous studies, trust in vaccines aids in explaining the acceptance of vaccination uptake [45]. In the current SARS-CoV-2 pandemic, development of new vaccination platforms has been achieved. Those surveyed in our study reported a high trust in all vaccines, albeit the CoronaVac and Pfizer vaccines proved to be the most trusted. While the CoronaVac has reported lower efficacies and effectiveness than Moderna and AstraZeneca [25,46], it has received more trust among Chilean residents. The high trust in the Pfizer and CoronaVac vaccines coincides with them being the most administered in the country. Governments must proactively provide information on selected vaccine manufacturers and new platforms to break down potential knowledge barriers that may affect trust in vaccines. Consistent with other studies [12,29], our results strengthen the idea that high trust in vaccines and a high perceived effectiveness of prevention practices increase the willingness of people to accept vaccination for themselves and their children, as well as to accept a booster dose and annual vaccination programs.

Our results emphasize that the preoccupation regarding side effects of vaccines decreased the acceptance of SARS-CoV-2 vaccines [29,45,47]. One of our novel findings in the existing literature is that the preoccupation regarding the side effects of vaccines also decreased the willingness to vaccinate children, while it did not affect the willingness to accept a booster dose and annual vaccination programs. We also found that determinants of willingness to accept vaccination vary among genders and cohorts. For instance, trust in scientists and medical professionals influenced the willingness to accept SARS-CoV-2 vaccines and a booster dose only among men and adults, but not among women or young individuals. The reasons for this gender difference are not clear. They could be explained by trust; however, this is not enough to explain women’s willingness to be vaccinated, which should be further explored in future research.

Our findings must be considered in light of some limitations. First, we captured different degrees of confidence through an ordinal scale with four values, while the variability of confidence may be higher. Second, although we included different socio-demographic characteristics, additional aspects should be included, such as cultural, ethnic, rurality and income, among others. Third, provided the context of physical distance, the application of the online survey may have excluded people with less accessibility or knowledge regarding the Internet and technology. Despite these limitations, the present study is consistent and extremely relevant, clearly identifying the underlying factors affecting the willingness to accept a SARS-CoV-2 vaccine, an annual vaccination, booster doses, and the vaccination of children.

This study provides fundamental information regarding aspects such as trust and confidence toward key factors and stakeholders involved in the COVID-19 vaccination process. Other aspects, such as communication strategies, social media, information needs, or interlocutors must be further investigated in future qualitative studies. It could also be relevant to identify and differentiate between the perception of risk in relation to COVID-19 and knowledge of this virus. The widespread acceptance of the Chilean vaccination program against COVID-19 may be useful to establish future strategies, information pathways, and vaccination processes that increase the acceptance of future vaccines in Chile and around the world.

## 5. Conclusions

In conclusion, our study reports that Chilean people show a high intention to receive a booster dose of a COVID-19 vaccine, an annual vaccination, and vaccination of children under 16 years of age. Vaccine acceptance is associated with trust in scientists, medical professionals, and in different media, such as television and radio. The perceived risk of becoming ill with COVID-19 is a factor that determined the acceptance of vaccination, childhood vaccination, and booster doses in adults.

## Figures and Tables

**Figure 1 vaccines-10-00681-f001:**
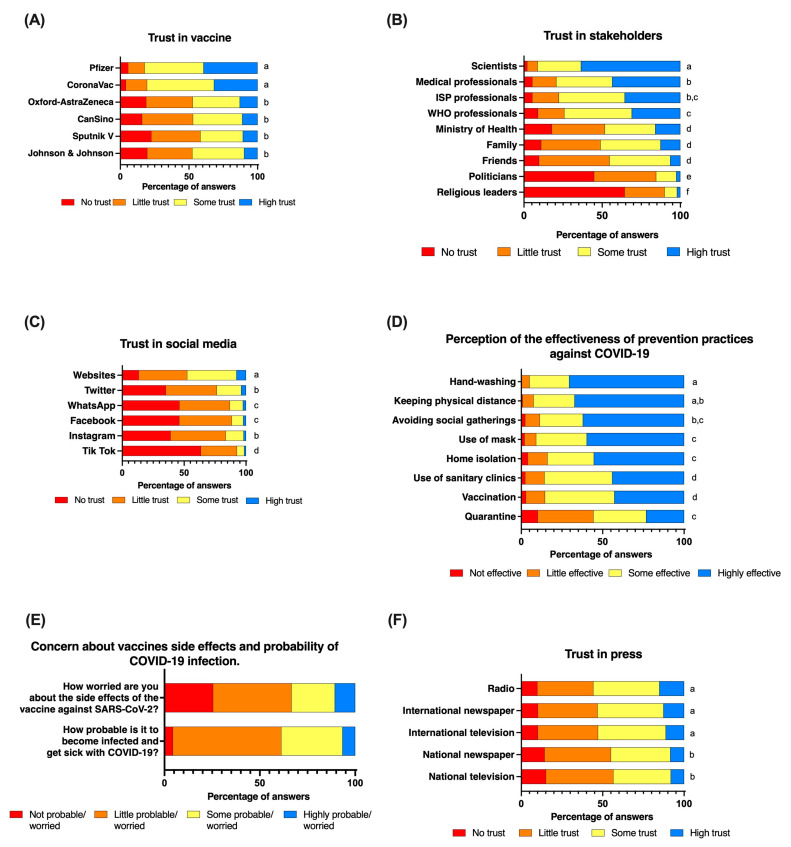
Respondents’ perceptions of vaccine side effects, getting sick from COVID-19, effectiveness of COVID-19 control practices, and trust in vaccines, stakeholders, social media, and the press (*n* = 744). The bars indicate the breakdown of the percentage of respondents providing an answer to each question asked. The full questionnaire is shown in the Appendix A. ISP (Chilean Public Health Institute), WHO (World Health Organization). (**A**) Trust in vaccine. (**B**) Trust in stakeholders. (**C**) Trust in social media. (**D**) Perception of the effectiveness of prevention practices against COVID-19. (**E**) Concern about vaccines’ side effects and probability of COVID-19 infection. (**F**) Trust in the press. For all figures, different lowercase letters indicate a significant difference of trust or perception (Dunn’s test with Bonferroni adjustment, *p* < 0.01).

**Table 1 vaccines-10-00681-t001:** Associations regarding the willingness to accept a SARS-CoV-2 vaccination, third dose, annual vaccination, and the vaccination of children, with variables of trust and perception among Chileans (*n* = 744).

		Outcome Variables			
Explanatory Variables		Willingness to Receive a SARS-CoV-2 Vaccination	Willingness to Receive a Third Dose Vaccination	Willingness to Receive an Annual Vaccination	Willingness to Vaccinate Children
		(1)	(2)	(3)	(4)
Trust in vaccines	(a)	4.1 ** (2.0–8.2)	3.2 ** (1.8–5.6)	2.1 ** (1.6–2.8)	1.9 ** (1.4–2.6)
Trust in scientists and medical professionals	(b)	2.4 * (1.2–5.0)	2.8 ** (1.5–5.0)	2.2 ** (1.6–3.1)	2.6 ** (1.8–3.6)
Trust in politicians	(c)	2.5 * (1.1–5.6)	1.5 (0.8–2.6)	1.2 (0.9–1.6)	1.3 (0.9–1.7)
Trust in religious leaders	(d)	0.9 (0.5–1.8)	0.7 (0.4–1.0)	0.7 * (0.6–1.0)	0.7 ** (0.5–0.9)
Trust in relatives	(e)	1.7 (0.9–3.3)	1.3 (0.8–2.1)	1.1 (0.8–1.4)	1.2 (0.9–1.6)
Trust in social media	(f)	0.9 (0.4–1.9)	0.4 ** (0.2–0.7)	1.0 (0.7–1.3)	0.7 * (0.5–1.0)
Trust in press	(g)	0.8 (0.3–1.7)	1.4 (0.8–2.6)	1.1 (0.8–1.5)	1.1 (0.8–1.6)
Perceived effectiveness of prevention practices	(h)	2.1 * (1.0–4.5)	2.4 ** (1.3–4.5)	2.4 ** (1.6–3.4)	2.4 ** (1.6–3.5)
Perceived risk of infection	(i)	2.0 * (1.1–3.7)	1.5 (0.9–2.3)	1.4 * (1.1–1.8)	1.2 (0.9–1.5)
Preoccupation regarding side effects of vaccines	(j)	0.6 ** (0.4–0.9)	0.9 (0.6–1.2)	0.9 (0.8–1.1)	0.8 * (0.7–1.0)
Perceived comprehension of vaccines	(k)	0.7 (0.4–1.2)	0.6 (0.4–1.0)	1.1 (0.8–1.4)	1.1 (0.8–1.4)
Perceived prevention of severity due to vaccines	(l)	1.3 (0.9–1.9)	1.0 (0.7–1.3)	1.0 (0.8–1.1)	0.8 ** (0.7–0.9)
Perceived relaxation of prevention practices thanks to vaccination	(m)	1.4 (0.8–2.3)	0.7 (0.5–1.1)	0.7 ** (0.6–0.9)	1.0 (0.8–1.3)
Perceived possibility of the vaccination stopping the pandemic	(n)	1.3 (0.9–2.0)	1.4 * (1.0–1.9)	1.4 ** (1.2–1.7)	1.3 ** (1.1–1.5)
Perceived impact on quality of life	(o)	0.6 * (0.3–1.0)	1.0 (0.6–1.6)	0.9 (0.7–1.1)	0.8 (0.6–1.1)
COVID-19 infection in family	(p)	0.7 (0.3–1.7)	0.9 (0.5–1.7)	1.0 (0.7–1.4)	0.8 (0.5–1.1)
Age	(q)	1.1 * (1.0–1.1)	1.0 (1.0–1.0)	1.0 (1.0–1.0)	1.0 ** (1.0–1.1)
Gender (Women = 1)	(r)	1.3 (0.5–3.2)	0.9 (0.4–1.7)	1.1 (0.8–1.6)	1.3 (0.9–1.9)
Schooling	(s)	0.9 (0.7–1.2)	1.0 (0.8–1.2)	0.9 (0.8–1.0)	0.9 (0.8–1.0)
Multivariate model		Ordered logit	Logit	Ordered logit	Ordered logit

Columns (1), (3), and (4) show the results of the ordered logit multivariate models. Column (2) shows logit model results. For all columns, cells show odd ratio coefficients and, in parenthesis, confidence intervals at 95%. For each outcome variable, Table 1 shows the model with the best goodness of fit and parsimony compared to other candidate models, which was selected using Akaike Information Criterion (see Appendix A). * and ** refer to significant levels at 5% and 1%.

## Data Availability

The data sets generated and/or analyzed during the current study are available from the corresponding authors upon reasonable request.

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
