# Peer review of "Factors Influencing the Acceptance of COVID-19 Vaccines in a Country with a High Vaccination Rate"

_vaccines, 2022, doi:10.3390/vaccines10050681_

Round 1

Reviewer 1 Report

The manuscript with manuscript ID: vaccines-1656288 entitled "Factors influencing the acceptance of COVID-19 vaccines in a country with a high vaccination rate" is seems good research. The manuscript is well written and has sufficient data and may be published in the Vaccines after major revision. I have many points for the authors to consider:

  • However, this paper is good, self-explanatory and can be acceptable with revision after typographical and grammatical corrections.
  • The manuscript is on the whole well written but there are some problems with the English (including tenses, plurals, matching of adjectives and nouns, adverbs and verbs) such that other sections are almost ‘good’ and I would be inclined to rectify the partial contribution for plagiarism.
  • In introduction authors indicated that “By July 2021, Chile was among the first countries in the world with more doses administered per 100 people”. Kindly recheck and supports with more references.
  • Material and methods What was the reason to select the time frame June and July 2021 and I think that sample size is low to predict any conclusive result.
  • For the better analysis, if possible, the authors are advised to include three categories of population,
  1. A pocket from general population where very least population were vaccinated
  2. A pocket from general population where moderate population were vaccinated
  • A pocket from general population where maximum population were vaccinated

I think that this whole study lacks the above control from general population as the study.

  • In Discussion, it is stated that “Our results indicate that higher acceptance of SARS-CoV-2 vaccines, a booster dose, 446 annual vaccination and vaccination of children correlates with a high level of trust in experts in the field (scientists and medical professionals). These significant and positive associations emphasize the pivotal role that trust in experts plays in the vaccination process against COVID-19. In contrast, trust in political or religious leaders proved to be extremely low and, on the contrary, when it was high, the refusal to vaccinate was also higher. Our results are consistent with other studies, where a high trust in health workers has been associated with high acceptance of vaccination”. Kindly rewrite the above and support the above with many references. However, I think that for the successful vaccination strategy, the role of government lead by politician is crucial followed by religious leaders.
  • I also feel that authors should also try to focus on the Government strategy to boost up the vaccination.
  • If the authors make the suggested changes, then accept this article because being as important contribution on factors influencing the acceptance of COVID-19 vaccines. If they do not then the article will be yet another of those ‘tantalizing’ article that promises much but deliver little and therefore do not get cited.

Author Response

Dear Reviewer, 

Thank you for giving us the opportunity to submit a revised draft of our manuscript titled “Factors influencing the acceptance of COVID-19 vaccines in a country with a high vaccination rate” to Vaccines. We appreciate the time and effort that you have dedicated to providing your valuable feedback regarding our manuscript. Here is a point-by-point response to your comments and concerns.

Best wishes,

Reviewer 2 Report

The paper deals with an important question, namely the factors that influence the acceptability of COVID-19 vaccines.

The authors explored a number of factors and provide interesting results. The work looks relevant and publishable. I have several comments that could help:

  • Logistic regression is used to identify variables that significantly affect the willingness to accept vaccination. However, it is not completely clear to me what variables are included for each of the studied outcome variables. Are all variables (183?) included in all cases? If so, I am not sure if corrections for multiple testing were implemented. Perhaps a table making more explicit the details of Eq. (1) used in each case would help clarify this point.
  • p 4, l 183: “alpha, beta and gamma represent the odds ratio as a measure of association among 183 variables.” I have two questions on this:
    • Strictly speaking, I would say that the odds rations are the exponential of these regression coefficients.
    • This is probably related to my previous comment on the interpretation of Eq. (1): It looks mathematically imprecise to mention three coefficients alpha, beta and gamma and then say that there are 183 variables. How are the 183 variables distributed among the alpha, beta and gamma?
  • It could be that some of the co-variates are actually related to each other and this might affect the inferred conclusions. I understand that this might require significant additional work but was wondering if this was considered at all in case it is of relevance.
  • Most of the initial paragraphs in section 3 (before 3.1) look like methods. They look misplaced.
  • “Cronbach’s Alpha” and other statistical terminology might not be clear to some readers. I suggest adding references to literature where key concepts are explained. Also, the level of detail does not look balanced at places. For instance, the concept of odds ratios is explained in some detail on page 5 but other potentially challenging concepts are not explained. I am not necessarily asking the authors to explain everything in detail but just to keep a balance.

Author Response

(The authors gave the same response as above.)
